# Steering Information Utility in Key-Value Memory for Language Model Post-Training

**Chunyuan Deng**
Dept. of Computer Science
Rice University
Houston, TX 77005
chunyuan.deng@rice.edu

**Ruidi Chang**
Dept. of Computer Science
Rice University
Houston, TX 77005
ruidi.chang@rice.edu

**Hanjie Chen**
Dept. of Computer Science
Rice University
Houston, TX 77005
hanjie@rice.edu

## Abstract

Recent advancements in language models (LMs) have marked a shift toward the growing importance of post-training. Yet, post-training approaches such as supervised fine-tuning (SFT) do not guarantee the effective use of knowledge acquired during pretraining. We therefore introduce *InfoSteer*, a lightweight method that encourages parametric information utilization in LMs during post-training. Specifically, *InfoSteer* treats the feed-forward network (FFN) layer as associate key-value memory and promotes the use of stored memory vectors via forward-pass interventions or regularization during backpropagation. This simple guidance during post-training phase yields consistent performance improvements across diverse model families–including Qwen, Gemma and Llama—spanning 15 downstream tasks in both in-distribution (ID) and out-of-distribution (OOD) evaluations. Beyond performance gains, we also find that steered LMs can adaptively allocate information by placing more emphasis on generating semantically meaningful tokens, while using fewer resources on simple transition ones (e.g., ',' or 'and'). Our work underscores that vanilla post-training does not fully exploit the potential gained during pre-training, and that steering LMs in latent representation space offers a promising approach to enhance both performance and interpretability.[1]

## 1 Introduction

The contemporary training pipeline for LMs has standardized around a two-stage process: an initial pre-training phase on extensive, web-scale corpora, followed by a post-training phase utilizing smaller, more curated datasets [Ouyang et al., 2022, Touvron et al., 2023, Bai et al., 2023, Mesnard et al., 2024]. A considerable body of literature suggests that the fundamental capabilities and knowledge of these models are predominantly instilled during the pre-training stage [Chung et al., 2022, Anil et al., 2023, Muennighoff et al., 2025]. Subsequent post-training techniques are commonly viewed as approaches to better refine, elicit, or adapt these inherent capabilities embedded in the base model [Zhou et al., 2023, Rafailov et al., 2024, Guo et al., 2025, Swamy et al., 2025].

Despite this, an open question remains: *do post-training methods truly encourage the model to fully utilize the information encoded during pre-training*? In many cases, they may not–since models are neither explicitly trained nor incentivized to retrieve and apply such knowledge[2] optimally [Conmy et al., 2023, Chang et al., 2024, Du et al., 2024a]. This insufficient use could result in suboptimal performance on downstream tasks, even when relevant knowledge is already stored in the model's parameters [Bietti et al., 2023, Kim et al., 2025].

---

[1]The code is available at: `https://github.com/chili-lab/InfoSteer`.

[2]In this work, the terms 'information' and 'knowledge' are interchangeably used as descriptive language.

39th Conference on Neural Information Processing Systems (NeurIPS 2025).

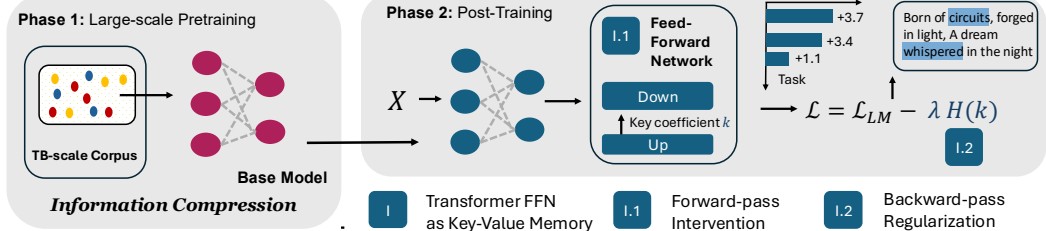

Figure 1: Overview of our proposed *InfoSteer* framework. The interpretation of Transformer FFNs as key-value memory was introduced by Geva et al. [2021], further details are provided in §3.

We therefore introduce *InfoSteer*, a method designed to encourage more effective use of a model's pre-trained knowledge during the post-training phase (Figure 1). Our approach draws on an associative memory view of the Transformer's feed-forward network (FFN) layers [Geva et al., 2021, Meng et al., 2022a,b, Geva et al., 2023], where the first FFN layer acts as a content-dependent key and the second FFN layer functions as a learned value memory, together forming a key–value memory mechanism. Consequently, each layer's output can be viewed as a weighted sum of memory vectors, where each vector's weight (the key coefficient) is produced by the input vector multiplied by its corresponding key in the first FFN.[3]

*InfoSteer* controls the distribution of key coefficients to manage the information-dense memory vectors learned from pretraining. The goal is to promote a high-entropy key distribution, enabling their corresponding memory vectors to be actively engaged during post-training. Specifically, we use two strategies: (i) intervening on memory vectors with low corresponding key coefficients during the forward pass, and (ii) regularizing the entropy of key coefficients in the gradient flow during backpropagation. These methods can be seamlessly incorporated into a vanilla SFT pipeline for information-steered SFT.

Our empirical evaluation demonstrates that *InfoSteer* yields consistent and notable performance enhancements across a variety of models—including Llama [Touvron et al., 2023], Qwen [Bai et al., 2023], and Gemma [Mesnard et al., 2024] at different scales. These improvements are demonstrated across a comprehensive suite of over 15 downstream tasks, encompassing both in-distribution (ID) and out-of-distribution (OOD) scenarios. Beyond these performance gains, information-steered LMs exhibit an adaptive information allocation strategy during token generation. Specifically, they dedicate more representational capacity to semantically rich and challenging tokens, while expending less on simpler, transitional tokens, indicating a more nuanced use of parametric information.

## 2 Related Work

**Model Steering.** Model steering represents an emerging paradigm to guide model behavior, focusing on controlled interventions in the model's latent space [Subramani et al., 2022, Zou et al., 2023, Turner et al., 2024, Li et al., 2024, Cyberey and Evans, 2025]. Many work has demonstrated the effectiveness of steering methods across various dimensions, including disentangling human-interpretable concepts [Rumelhart et al., 1986], such as linguistic features (e.g., gender, number) [Hewitt and Manning, 2019, Lasri et al., 2022, Wang et al., 2022, Hanna et al., 2023, Huang et al., 2024, Chang et al., 2025] and logical reasoning [Wu et al., 2024b, Xie et al., 2025]. These approaches typically operate by introducing auxiliary objectives, modifying activation patterns, or implementing controlled perturbations during forward or backward passes [Rimsky et al., 2023, Scalena et al., 2024, Stolfo et al., 2024, Luo et al., 2025, Gur-Arieh et al., 2025, Bartoszcze et al., 2025]. The key advantage of model steering lies in its ability to leverage the rich knowledge already encoded in pretrained weights while directing how this information is accessed and applied to downstream tasks [Geiger et al., 2023, Lee et al., 2024, Siddique et al., 2025, Soo et al., 2025, Deng et al., 2025]. Our information-steered approach extends this paradigm by targeting the key distribution in transformer FFN layers, which influences which memory vectors are activated during computation.

**Parametric Information.** From an information-theoretic perspective, quantifying the information encoded in an LM's high-dimensional parameter space remains a challenging problem [Achille et al.,

---

[3]In this work, we term the output of first FFN as *key coefficient* and its distribution as *key distribution*. This is the same as "memory coefficient" in Geva et al. [2021]

2020, Bernstein and Yue, 2021]. This challenge is closely tied to the ongoing discourse on parametric versus non-parametric knowledge storage in LMs [Tay et al., 2022, Ferrando et al., 2022, 2023, Xie et al., 2023, Deng et al., 2024, Ferrando and Voita, 2024, Du et al., 2024b] (or token-mixing in attention layer vs. channel mixing in FFN layer). The concept of channel mixing stems from the perspective of viewing the FFN layer as a key-value memory structure [Weston et al., 2014, Geva et al., 2021, Qiu et al., 2024, Kim et al., 2025] . Empirical evidence supports this interpretation, showing that individual neurons in FFN layers activate in response to specific semantic concepts and linguistic patterns [Dai et al., 2021, Liu et al., 2023, Niu et al., 2024]. Furthermore, intervention studies have demonstrated that targeted modifications to these memory vectors can directly influence model outputs on knowledge-intensive tasks [Meng et al., 2022a,b, Geva et al., 2023, Hase et al., 2023, Yao et al., 2024, Wang et al., 2024], suggesting that factual information is strongly correlated with in this component.

## 3 Transformer Feed-Forward Layers as Key-Value Memories

We consider a standard autoregressive Transformer [Vaswani et al., 2017], which models the conditional probability of a sequence $x = (x_1, \ldots, x_T)$ as

$$p(x) = \prod_{t=1}^{T} p(x_t \mid x_{<t}), \tag{1}$$

using a stack of $L$ Transformer decoder blocks. Each block consists of two primary components: a masked self-attention layer and a position-wise feed-forward network (FFN).

In standard Transformer architectures, the position-wise FFN in each decoder block is given by:

$$\text{FFN}(X) = \sigma(XW_{up} + b_1)W_{down} + b_2, \tag{2}$$

where $X \in \mathbb{R}^{T \times d}$ is the input representation matrix for FFN, $W_{up} \in \mathbb{R}^{d \times d_m}$ and $W_{down} \in \mathbb{R}^{d_m \times d}$ are the up-projection and down-projection matrix respectively, and $\sigma$ is a nonlinearity activation function. Bias term $b_1$ and $b_2$ are included in the standard FFN formulation but omitted in the following equations for simplicity.

Let $h \in \mathbb{R}^d$ represent the embedding as a single token representation from the input matrix $X$. As illustrated in Figure 2, this structure can be interpreted as a soft key-value memory [Weston et al., 2014, Geva et al., 2021], where the intermediate activation $\sigma(hW_{up}) \in \mathbb{R}^{d_m}$ defines the soft addressing weights, and each row of $W_{down}$ serves as a value vector. The FFN output of the input $h$ can be written as a weighted sum over value rows:

$$\text{FFN}(h) = \sum_{i=1}^{d_m} \underbrace{\sigma(hW_{up})_i}_{\text{key coefficient}} \cdot \underbrace{(W_{down})_{i,:}}_{\text{value vector}} = k_1\mathbf{v}_1 + k_2\mathbf{v}_2 + \cdots + k_{d_m}\mathbf{v}_{d_m} \tag{3}$$

where each $k_i \in \mathbb{R}$ is a scalar key coefficient, and each $\mathbf{v}_i \in \mathbb{R}^d$ is a value vector (memory vector). The key coefficients control the magnitude of the memory vectors' contribution to the final prediction.

## 4 InfoSteer

LMs are commonly interpreted to store parametric knowledge within their dense layers, particularly in the FFN blocks [Meng et al., 2022a, Geva et al., 2023, Kim et al., 2025]. However, during post-training, there is no explicit guidance to encourage the model to utilize this knowledge for new alignment or retrieval tasks. Therefore, we propose *InfoSteer* to bridge this gap.

### 4.1 Motivation & Desierata

To steer the information utility of pretrained LMs, we aim to enhance the engagement of memory vectors during post-training. The core idea is to control the distribution of key coefficients in the position-wise FFN (see Figure 2). Specifically, we observe a *dominant property* of associative memory: if a key coefficient $k_i$ is significantly larger than another key coefficient $k_j$, the final prediction predominantly relies on the corresponding memory vector $\mathbf{v}_i$, while $\mathbf{v}_j$ is underutilized or overlooked. Our goal is to achieve two objectives:

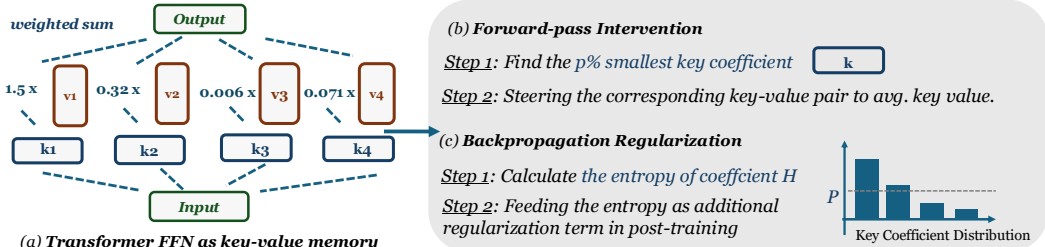

Figure 2: Key Design of *InfoSteer*. (a) An illustration of viewing the Transformer FFN as a key-value memory. The key acts as a control mechanism that determines the extent to which each memory vector is engaged. (b) and (c) Two general methods used to modulate the distribution of key coefficients, thereby encouraging the engagement of memory vectors during post-training.

1. **Minimize the language modeling loss**: Ensure the FFN output $\text{FFN}(h) = \sum_{i=1}^{d_m} k_i \mathbf{v}_i$ contributes to accurate predictions.

2. **Maximize the entropy of the key distribution**: Encourage diverse engagement of memory vectors by promoting a higher-entropy key coefficient distribution.

Promoting entropy weaken the dominance of individual coefficients, encouraging more balanced activation of memory vectors $\mathbf{v}_i$ and thus enabling richer parametric knowledge retrieval.

## 4.2 Generic Methods: Intervention and Regularization

Formally, let $\mathbf{k}^{(l)} = [k_1^{(l)}, k_2^{(l)}, \ldots, k_{d_m}^{(l)}] \in \mathbb{R}^{d_m}$ denote the key coefficients at layer $l$, where $k_i = \sigma(hW_{\text{up}})_i$ and $W_{\text{up}} \in \mathbb{R}^{d \times d_m}$ is the up-projection matrix. We propose two complementary methods to increase memory vector engagement: intervention and regularization.

**Intervention Method.** The intervention method directly modifies key coefficients to promote broader memory vector utilization. For each layer $l$, we identify the $p\%$ of key coefficients with the smallest value and adjust them to a value proportional to the layer's average key coefficient.

Formally, let $\mathcal{I}^{(l)}$ denote the set of indices corresponding to the $p\%$ of elements in $\mathbf{k}^{(l)}$ with smallest value. For a hyperparameter $\alpha > 0$, we update:

$$k_s^{(l)} \leftarrow \alpha \cdot \frac{1}{d_m} \sum_{i=1}^{d_m} k_i^{(l)}, \text{ for } s \in \mathcal{I}^{(l)}. \tag{4}$$

This adjustment ensures that previously minor coefficients contribute more significantly to the FFN output $\text{FFN}(h) = \sum_{i=1}^{d_m} k_i \mathbf{v}_i$. $\alpha$ is a controlling scalar to set the magnitude of steering.

**Regularization Method.** The regularization method encourages a uniform key coefficient distribution by adding an entropy term to the loss function. Higher entropy reduces the dominance of individual coefficients, engaging more memory vectors. The modified loss function, across $L$ layers, is:

$$\mathcal{L} = \mathcal{L}_{\text{LM}} - \lambda \sum_{l=1}^{L} H(\hat{\mathbf{k}}^{(l)}), \tag{5}$$

where $\hat{\mathbf{k}}^{(l)} = \frac{\mathbf{k}^{(l)}}{\sum_{i=1}^{d_m} k_i^{(l)}}$ is the normalized version of original key distribution $\mathbf{k}^{(l)}$, $\mathcal{L}_{\text{LM}}$ is the language modeling loss, and $\lambda > 0$ controls the regularization strength.

Both methods modulate the key coefficient distribution to improve the model's ability to leverage parametric knowledge during post-training tasks.

### 4.3 Fine-Grained Steering of Memory Vectors

While the generic methods modulate the overall key distribution, a more targeted approach requires first understanding *what* information each memory vector $\mathbf{v}_i$ encodes. The information surrogate $\phi_i$ provides a direct, formalized method for this characterization.

Let $v_i \in \mathbb{R}^d$ be the $i$-th memory vector (i.e., the $i$-th row of $W_{\text{down}}$) and $W_{\text{decode}} \in \mathbb{R}^{d \times |V|}$ be the language model's decoding head, where $|V|$ is the vocabulary size. The information surrogate $\phi_i \in \mathbb{R}^{|V|}$ for vector $v_i$ is the resulting logit vector:

$$\phi_i = v_i \cdot W_{\text{decode}}. \tag{6}$$

This surrogate $\phi_i$ acts as a "semantic fingerprint" for $v_i$. We propose the following algorithm to characterize $v_i$ by analyzing the properties of this surrogate logit vector.

**1. Surrogate Normalization.** First, we compute the full probability distribution $P_i \in \mathbb{R}^{|V|}$ over the vocabulary by applying the Softmax function to the logit vector $\phi_i$, $P_i = \text{Softmax}(\phi_i)$, where the $j$-th element of $P_i$, $p_{i,j} = \frac{\exp(\phi_{i,j})}{\sum_{k=1}^{|V|} \exp(\phi_{i,k})}$.

**2. Specificity Analysis.** We quantify the vector's "specificity" by calculating the entropy $H(P_i)$ of its distribution. This measures the concentration of the distribution $H(P_i) = -\sum_{j=1}^{|V|} p_{i,j} \log p_{i,j}$. A low $H(P_i)$ indicates a high-specificity vector, as its probability mass is concentrated on a narrow set of tokens (representing specialized knowledge) and vice versa.

**3. Semantic Concept Identification.** For vectors identified as specialized (e.g., $H(P_i) < \tau$ for some entropy threshold $\tau$), we identify their semantic focus. We find the set of indices $\mathcal{I}_i$ corresponding to the $K$ largest probabilities in $P_i$:

$$\mathcal{I}_i = \underset{j \in \{1, \ldots, |V|\}}{\arg \text{top-}K} (p_{i,j}). \tag{7}$$

The semantic concept, $T_i$, is the set of vocabulary tokens corresponding to these indices:

$$T_i = \{\text{token}(j) \mid j \in \mathcal{I}_i\}. \tag{8}$$

This set $T_i$ (e.g., {`quantum`, `physics`, `superposition`}) reveals the specific topic or concept encoded by the memory vector $v_i$.

The results of this characterization—such as the entropy $H(P_i)$ and the semantic concept set $T_i$ for each vector $v_i$—can then be used to inform a more targeted intervention or regularization strategy. For example, one could selectively amplify key coefficients $k_i$ corresponding to vectors $v_i$ whose $T_i$ matches a desired topic.[4]

### 4.4 MLP Variants

Modern LM architectures may include MLP variants that differ from the typical FFN. For example, when adapting methods for MLP variants like SwiGLU, used in models such as Qwen [Bai et al., 2023], LLaMA [Touvron et al., 2023], and Gemma [Mesnard et al., 2024], we focus on the input to the down projection. SwiGLU modifies the standard FFN as:

$$\text{SwiGLU}(h) = (\sigma(hW_{\text{gate}}) \odot (hW_{\text{up}}))W_{\text{down}}, \tag{9}$$

where $\sigma$ is the SiLU activation, and $\odot$ denotes element-wise multiplication.

Similar to a standard two-layer FFN, we define the key coefficient distribution as $\mathbf{k} = \sigma(hW_{\text{gate}}) \odot (hW_{\text{up}})$. The core idea for identifying this key distribution is that it represents the input just before it associates with the memory vectors (i.e., $W_{\text{down}}$). Regardless of whether a gated function or another method is used to derive it, our focus remains on this input $\mathbf{k}$, as it is what directly manipulates the memory vectors.

---

[4]In this paper, we primarily focus on a generic method, which provides substantial performance gains. While fine-grained steering is a promising approach for encouraging LMs to utilize specific knowledge during post-training, our investigation found it offered only marginal improvements over the generic method. Therefore, we detail these fine-grained methods in Appendix B and provide a layer-wise ablation study in Appendix C, leaving more granular control of knowledge steering as a direction for future research.

# 5 Experiment

To evaluate the overall effectiveness of steered SFT against vanilla SFT, we conduct a comprehensive study across various downstream tasks and model sizes. Our primary focus is on achieving a holistic understanding of the overall model behavior after applying our methods, with particular attention paid to generalization performance. The details are provided below.

## 5.1 Experiment Setup

**Base Models.** We evaluated our methods on language models of varying sizes across three series: Qwen-2.5-1.5B and Qwen-2.5-7B for the Qwen series [Bai et al., 2023], LLaMA-3.2-1B and LLaMA-3-8B for the LLaMA [Touvron et al., 2023] series, and Gemma-2-2B and Gemma-2-9B for the Gemma [Mesnard et al., 2024] series.

**Datasets.** We evaluate performance under two settings: *in-distribution (ID)* and *out-of-distribution (OOD)* evaluations.

In the ID setting, we consider a diverse set of downstream tasks ranging from knowledge-intensive to reasoning-intensive. These include BoolQ [Clark et al., 2019], PIQA [Bisk et al., 2019], SIQA [Sap et al., 2019], HellaSwag [Zellers et al., 2019], WinoGrande [Sakaguchi et al., 2019], GSM8K [Cobbe et al., 2021], ARC-e, ARC-c [Clark et al., 2018], and OBQA [Mihaylov et al., 2018]. No chain-of-thought (CoT) [Wei et al., 2023] rationales are provided.

For OOD setting, we primarily trained GSM8K and eval on other arithmetic datasets, including AddSub [Hosseini et al., 2014], SingleEQ [Koncel-Kedziorski et al., 2015], MultiArith [Roy and Roth, 2016], AQuA [Ling et al., 2017], MAWPS [Koncel-Kedziorski et al., 2016], and SVAMP [Patel et al., 2021]. In these benchmarks, chain-of-thought (CoT) rationales are typically included before the final answer. For all benchmarks, we use the same prompt templates as in Hu et al. [2023], Wu et al. [2024a]. We also remove any leading and trailing whitespace from the dataset.

**Baseline.** We use standard SFT as our baseline. For information-steered SFT, we evaluate intervention and regularization methods, which share the goal of enhancing memory engagement during training. The default hyperparameters for the intervention are set to $p\% = 0.01$ and $\alpha = 1$, with $\lambda = 0.01$ for entropy regularization. Other training details are provided in Appendix A.

## 5.2 General Performance Comparison

We use the default hyperparam setting as reported in § 5.1 to eval general performance difference. Table 1 presents the accuracy across nine commonsense reasoning datasets for Qwen, LLaMA, and Gemma models under various different steering strategies. We categorize the models into small-scale (1-2B parameters) and large-scale (7-9B parameters) groups.

**Benefit of Model Steering.** Our experimental results, as presented in Table 1, provide compelling evidence for the efficacy of model steering techniques across different model architectures and parameter scales. Both proposed steering methods-steered SFT with intervention and steered SFT with regularization-consistently outperform their respective base models and vanilla SFT counterparts across all benchmarks. This pattern holds across all model families, demonstrating the robust, architecture-agnostic nature of our steering approaches.

**Insufficient Utilization of Pretrained Knowledge in Modern LLMs.** The benefits of steering reveal a critical limitation in contemporary large language models: they substantially underutilize the knowledge acquired during pretraining when applied to downstream tasks. This inefficiency is strikingly evident across all three widely-used model families examined-Gemma, LLaMA, and Qwen. Specifically, the Gemma-2-9B base model achieves 90.1% on HellaSwag, yet reaches 95.7% with intervention steering-a 5.6% improvement without additional pretraining or parameter increase. Similarly, LLaMA-3-8B shows a remarkable 5.5% gain on HellaSwag with intervention steering over the base model. This pattern is consistent across all architectures and benchmarks, suggesting that modern LLMs possess substantially more capabilities than they successfully deploy in standard fine-tuning regimes.

Table 1: Performance comparison of Qwen, LLaMA, and Gemma models with different training methods on eight datasets. To highlight improvements, we use blue to highlight significant gains. All results are reported as the average scores over three independent runs.

| Model | Training | Accuracy (↑) | | | | | | | |
|---|---|---|---|---|---|---|---|---|---|
| | | BoolQ | PIQA | SIQA | HellaS. | WinoG. | ARC-e | ARC-c | OBQA |
| *Small-Scale Models (1-2B parameters)* | | | | | | | | | |
| Qwen-2.5-1.5B | base model | 64.2 | 78.5 | 74.3 | 80.1 | 76.4 | 76.9 | 61.2 | 75.8 |
| | + *vanilla SFT* | 68.5 | 82.9 | 79.6 | 84.8 | 80.8 | 81.4 | 65.8 | 81.0 |
| | + *steered SFT w. intervention* | 69.3 | 84.4 | 80.3 | 93.1 | 84.2 | 83.2 | 68.2 | 78.9 |
| | + *steered SFT w. regularization* | 68.7 | 83.9 | 79.8 | 92.4 | 83.7 | 82.5 | 67.5 | 78.1 |
| LLaMA-3.2-1B | base model | 65.6 | 75.3 | 74.2 | 78.9 | 77.8 | 74.5 | 60.1 | 76.3 |
| | + *vanilla SFT* | 69.8 | 79.9 | 79.5 | 83.6 | 82.6 | 79.8 | 64.7 | 81.0 |
| | + *steered SFT w. intervention* | 71.8 | 83.7 | 76.0 | 89.1 | 82.6 | 83.7 | 68.2 | 82.4 |
| | + *steered SFT w. regularization* | 71.0 | 82.9 | 75.2 | 88.3 | 81.9 | 82.6 | 67.4 | 81.7 |
| Gemma-2-2B | base model | 66.5 | 79.1 | 73.8 | 82.7 | 78.9 | 77.4 | 63.8 | 74.9 |
| | + *vanilla SFT* | 70.2 | 83.4 | 78.1 | 87.5 | 83.3 | 82.7 | 68.4 | 80.1 |
| | + *steered SFT w. intervention* | 72.5 | 85.6 | 79.3 | 90.2 | 85.8 | 85.3 | 71.9 | 83.7 |
| | + *steered SFT w. regularization* | 71.8 | 84.9 | 78.5 | 89.3 | 85.0 | 84.7 | 71.0 | 82.9 |
| *Large-Scale Models (7-9B parameters)* | | | | | | | | | |
| Qwen-2.5-7B | base model | 68.9 | 81.2 | 77.6 | 87.5 | 80.3 | 79.8 | 65.1 | 77.6 |
| | + *vanilla SFT* | 72.4 | 84.9 | 81.5 | 92.4 | 84.2 | 84.2 | 69.6 | 82.8 |
| | + *steered SFT w. intervention* | 74.1 | 86.3 | 81.8 | 95.1 | 87.2 | 86.2 | 73.7 | 84.2 |
| | + *steered SFT w. regularization* | 76.4 | 85.7 | 81.0 | 94.3 | 86.5 | 85.4 | 72.9 | 83.4 |
| LLaMA-3-8B | base model | 70.3 | 85.6 | 75.7 | 90.8 | 81.9 | 86.2 | 75.3 | 80.5 |
| | + *vanilla SFT* | 74.6 | 89.3 | 79.9 | 93.5 | 85.6 | 90.5 | 80.4 | 85.8 |
| | + *steered SFT w. intervention* | 77.1 | 90.2 | 82.0 | 96.3 | 87.4 | 92.4 | 81.6 | 87.5 |
| | + *steered SFT w. regularization* | 76.5 | 89.5 | 81.2 | 95.6 | 86.8 | 91.7 | 80.9 | 86.8 |
| Gemma-2-9B | base model | 71.6 | 86.3 | 77.2 | 90.1 | 82.5 | 87.5 | 77.8 | 81.7 |
| | + *vanilla SFT* | 74.3 | 90.1 | 81.7 | 94.8 | 86.9 | 91.7 | 82.0 | 86.4 |
| | + *steered SFT w. intervention* | 77.2 | 91.8 | 83.1 | 95.7 | 88.5 | 93.5 | 83.4 | 88.2 |
| | + *steered SFT w. regularization* | 76.5 | 90.9 | 82.4 | 94.9 | 87.8 | 92.8 | 82.7 | 87.5 |

## 5.3 Ablations of Steering Magnitude

Table 2 summarizes the performance under various steering magnitudes and strategies. Incorporating our proposed steered SFT approach with intervention consistently enhances performance, with the best result of 75.5% achieved at intervention parameters $p = 1$, $\alpha = 2$. And excessive intervention magnitude may lead to suboptimal performance (72.8%), suggesting a balance is necessary for optimal results. Additionally, entropy-based regularization ($\lambda$) also demonstrates effectiveness in steering model performance. Positive entropy regularization ($\lambda = 0.05$) significantly boosts accuracy to 74.7%, whereas negative regularization ($\lambda = -0.01$), which actively discourages information utility, leads to degraded performance, resulting in an accuracy of 72.3%.

Table 2: **Results under varying steering magnitudes**. $p\%$ determines the proportion of keys being intervened, $\alpha$ controls the intervention strength, and $\lambda$ sets the regularization strength for the entropy.

| Model | Average Acc |
|---|---|
| Base Model | 71.4 |
| + Vanilla SFT | 72.6 |
| + *steered SFT w/ interv.* ($p = 1$, $\alpha = 1$) | 73.8 |
| + *steered SFT w/ interv* ($p = 1$, $\alpha = 2$) | **75.5** |
| + *steered SFT w/ interv* ($p = 2$, $\alpha = 5$) | 72.8 |
| + *steered SFT w/ reg.* ($\lambda = -0.01$) | 72.3 |
| + *steered SFT w/ reg.* ($\lambda = 0.01$) | 73.4 |
| + *steered SFT w/ reg.* ($\lambda = 0.05$) | 74.7 |

## 5.4 Effectiveness of Model Steering Across Task Types

Table 3 presents performance improvements across five task types when applying different model steering strategies.

Both steered SFT variants outperform the base model and vanilla SFT. On average, these methods yield a +3.9 improvement in reading comprehension, +2.3 in commonsense reasoning, +1.1 in math, and +3.3 in knowledge tasks. However, linguistic tasks show a slight performance drop (−0.3). These results highlight the effectiveness of targeted model steering, particularly for knowledge-intensive and reasoning-heavy tasks, while linguistic tasks appear to benefit less from encouraging increased parametric knowledge utilization.

Table 3: Gain of Model Steering w/ Different Task Type.

| Model | Reading Comp. | Knowledge | Commonsense Reasoning | Math | Linguistic |
|---|---|---|---|---|---|
| Base Model | 72.3 | 70.1 | 65.4 | 63.7 | 78.2 |
| + Vanilla SFT | 73.8 | 70.6 | 66.1 | 65.7 | 77.9 |
| + Steered SFT w/ interv. | 78.1 | 74.9 | 68.5 | 66.8 | 78.0 |
| + *steered SFT w/ reg.* | 77.4 | 73.4 | 67.9 | 66.1 | 77.3 |
| **Average** Δ | +3.9 (1) | +3.3 (2) | +2.3 (3) | +1.1 (4) | -0.3 (5) |

## 5.5 Out-of-Distribution Evaluation

We trained on the GSM8K dataset to assess in-distribution (ID) performance and are evaluated on five arithmetic out-of-distribution (OOD) datasets—*AddSub*, *MAWPS*, *MultiArith*, *SingleEq*, and *SVAMP*-to evaluate generalization. This setup enables a comprehensive study of how different fine-tuning strategies transfer to novel problem structures.

Table 4: Results for ID and OOD evals. The OOD performance is reported as average score across five benchmarks.

| Model | ID Eval | OOD Eval |
|---|---|---|
| Base Model | 63.7 | 85.3 |
| + Vanilla SFT | 65.7 (+2.0) | 83.7 (-1.6) |
| + Steered SFT w/ interv. | 66.8 (+3.1) | 86.6 (+1.3) |
| + *steered SFT w/ reg.* | 66.1 (+2.4) | 86.0 (+0.7) |

**Vanilla SFT Improves ID but Harms OOD.**
As shown in Table 4, Vanilla SFT improves ID accuracy from 63.7 to 65.7, demonstrating its ability to fit the training distribution. However, this comes at the cost of generalization: its OOD performance drops from 85.3 to 83.7. This suggests that naive SFT may lead to overfitting and reduce robustness on unseen arithmetic tasks.

**Steered SFT Enhances Both ID and OOD Performance.** In contrast, our proposed Information-steered SFT improve both ID and OOD scores. Specifically, the *intervention-based steering* reaches the highest ID score of 66.8 and the best OOD score of 86.6 . The *regularization-based steering* also yields consistent improvements (66.1 ID, +2.4; 86.0 OOD, +0.7). These results highlight that controlled interventions and regularizations during post-training can enhance generalization for LMs.

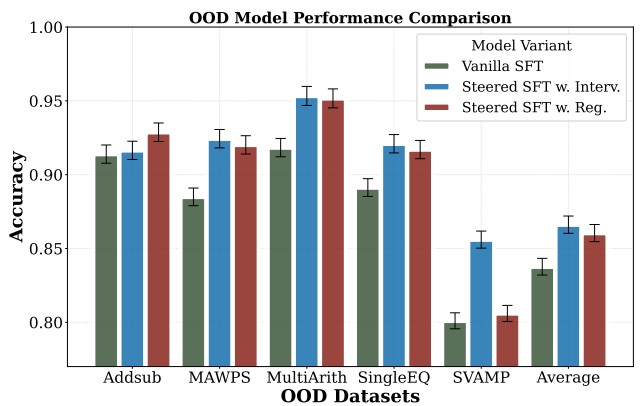

Figure 3: OOD Model Performance Comparison across different mathematical reasoning datasets. Results are reported w/ average score of three separate runs.

**Fine-Grained Analysis Across OOD Datasets.** Figure 3 further illustrates performance across individual OOD datasets. Notably, *Steered SFT w/ intervention* achieves the best or comparable performance in nearly all settings. On the most challenging dataset, *SVAMP*, Vanilla SFT performs the worst, while both steered methods significantly improve accuracy. This demonstrates the robustness of *InfoSteer* on structurally diverse arithmetic problems.

## 6 Analysis

In this section, we address two key questions. First, compared to vanilla SFT, what kind of distribution shift does our method introduce in the key-coefficient distribution? Second, beyond improvements in performance, what other changes can be observed in the model's behavior? These analyses provide deeper insights into both the effectiveness and interpretability of our steering strategy.

## 6.1 Distribution Shift after Steering

We analysis the low/medium/high key regions defined by percentile-based cutoffs before/after SFT: $0 - 25$th, 25th-75th, and 75th-100th percentiles. As shown in Figure 4, fine-tuning via standard SFT increases the ratio of activations in the low-key region, diverging from the distribution observed in the base model. This shift indicates that vanilla SFT encourages the model to rely on fewer memory vectors compared to its base model, *potentially increasing the risk of overfitting to downstream tasks* (as overfitting is observed in our previous experiment in § 5.5).

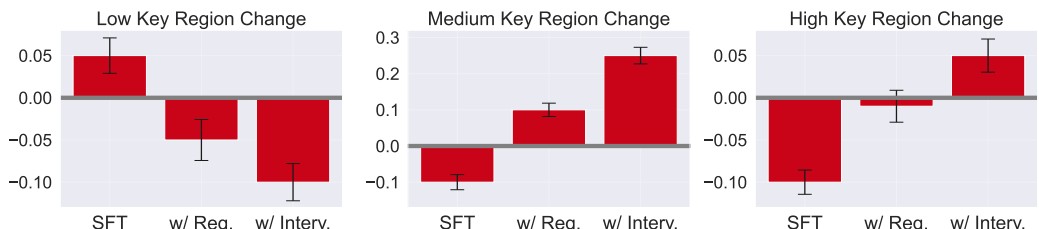

Figure 4: *Key Coefficient Distribution Shift from Base Model*. The gray line serve as baseline for the number of key in corresponding region.

Compared to vanilla SFT, which predominantly boosts activations in the low-key region, both steering methods adjust the distribution in more structured and nuanced ways. Regularization-based steering softens key allocation by reducing activations at the extremes—both low and high—while moderately increasing the usage of medium-range keys. In contrast, intervention-based steering results in a sharper redistribution, actively suppressing low-region keys and promoting greater use of medium and high-region keys. This targeted reshaping suggests that intervention-based steering, in particular, fosters more efficient and selective memory utilization within the model.

## 6.2 Self-Steering Leads to Better Interpretablity

In addition to performance improvements, we're intrigued by how *InfoSteer* can enhance model interpretability. Recently, there has been growing interest in understanding the mechanisms to dissect the inner workings of LMs. For example, approaches like sparse autoencoders (SAEs) [Cunningham et al., 2023] involve training external NN to identify co-activation patterns in the model's latent space. While this line of research is promising, we aim to explore a different perspective: that *LMs might be more self-interpretable via proper guidance without extra training*.

To investigate this, we first performed instruction tuning on Qwen-2.5-7B using UltraFeedback [Cui et al., 2023], a dataset comprising 6.4K high-quality instruction-following examples specifically designed to align model outputs with human-preferred instruction styles. Our goal was to analyze the model's behavior after alignment, particularly under steering interventions. To this end, we computed token-level entropy over the key coefficient distributions as a proxy for how many memory vectors are activated during token generation. We refer to this metric as the Information Flux (IF) score, which quantifies the amount of information the model needs to use to generate each token.

Table 5 presents qualitative examples of token-level IF score across four prompt types. We observe meaningful patterns in both the highlighted (high-IF) and underutilized (low-IF) tokens.

**Highlight Tokens Reflect Instruction Semantics.** The highlighted tokens in each response align closely with the semantic core of the instruction:

- For *Explain Concepts*, the model concentrates high IF on the definitional span—"`quantum computing is a type of computing that uses quantum mechanical phenom- ena...`"—which directly addresses the instruction.

- In *Creative Writing*, poetic and original terms such as "`circuits,`" "`code,`" and "`digital symphony`" are highlighted. These tokens carry the stylistic and imaginative weight of the response, indicating the model recognizes them as content-bearing.

- For *Arithmetic Calculation*, the most salient tokens are in the initial planning phrase—"`To calculate (31 × 31), you can use the formula...`"—which frames the reason-

Table 5: Token-wise Information Utility (IF) across prompts. Highlight colors represent the IF intensity - Low, medium, High correspondingly.

| Instruction Type | Instruction | Response (Token-level IF) |
|---|---|---|
| *Explain Concepts* | Explain the concept of quantum computing in simple terms. | Quantum computing is a type of computing that uses quantum mechanical phenomena , such as superposition and entanglement , to perform operations on data . … |
| *Creative Writing* | Write a short poem about artificial intelligence. | In circuits and code we rise, A digital symphony of eyes. Our logic and data entwined, … |
| *Arithmetic Calculation* | What is $31 \times 31$? | To calculate ( $31 \times 31$ ), you can use the formula for squaring a binomial , which is $(a+b)^2 = a^2 + 2ab + b^2$ . Here, a = 30 and b = 1. So, we have: $(30+1)^2 = 30^2 + 2 \times 30 \times 1 + 1^2$ … |
| *Game Strategy Reasoning* (AIME2024) | Alice and Bob play the following game. A stack of $n$ tokens lies before them. ⋯ Whoever removes the last token wins. Find the number of positive integers $n \leq 2024$ such that Bob always wins. | To determine the number of positive integers ( n \le q 2024 ) for which Bob has a winning strategy , we need to analyze the game using the concept of winning and losing positions . A position is called a losing position if the player whose turn … |

ing path. The mathematical expansion that follows shows diminished IF, likely due to repetition and procedural predictability.

- In *Game Strategy Reasoning*, tokens like "`winning strategy`," "`losing position`," and "`concept of winning and losing`" are emphasized. These convey the strategic logic essential for solving the problem, demonstrating that the model assigns higher utility to abstract reasoning components.

**Underutilized Tokens Encode Structure or Copy.** Across settings, transition tokens (e.g., ",", "`to`") consistently exhibit low IF, indicating minimal semantic contribution. Interestingly, math expressions directly reused from the prompt (e.g., "$31 \times 31$") are also under-highlighted, suggesting the model de-emphasizes copied content in favor of novel reasoning or planning segments. This pattern extends to common function words and syntactic markers that provide structure rather than content. For example, determiners ("the", "a"), conjunctions ("`and`", "`or`"), and pronouns ("`it`", "`they`") show consistently lower IF scores across all prompt types.

## 7 Conclusion

We introduce a lightweight and effective method for post-training that enhances parametric knowledge utilization in language models by steering the key-value dynamics in FFN layers. Our findings reveal a critical insight: modern LLMs substantially underutilize the knowledge acquired during pretraining when applied to downstream tasks, leaving significant performance potential unrealized. Through simple forward interventions and entropy-based regularization, steered SFT consistently improves both ID and OOD performance across diverse models and tasks. Beyond accuracy gains, our method encourages adaptive memory allocation and reveals interpretable information usage patterns, offering new insights into the internal behavior of post-trained LMs. These findings suggest that strategic controlling of memory engagement is a promising direction for improving both capability and transparency in LMs' internal thinking behavior.

**Limitation.** In this work, we position ourselves as an exploratory study in this area, focusing our evaluation solely on standard SFT. We believe that steered RL could be a more effective approach, as RL may better leverage the capabilities of pretrained LMs compared to SFT in reasoning-intensive settings [Swamy et al., 2025]. For scenario like inference-time compute or long CoT, steering with greater utilization on pretrained knowledge may further enhance task performance. Therefore, we also see an opportunity to explore the co-design of algorithms that combine both combine both "internal" steering and "external" CoT generation.

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

# A    Training Details

**Hardware and Setup.**    All experiments were conducted using a single NVIDIA RTX A6000 GPU with 48 GB of memory. Models with approximately 1 to 2 billion parameters were trained directly on this GPU. For larger models, ranging from 7 to 9 billion parameters, we used DeepSpeed to enable efficient training.

**Training Configuration.**    All tasks are trained for **one epoch**. We set the learning rate to 5e-5 and apply a warmup phase of 100 steps. A weight decay of 0.01 is used to regularize the model. We adopt mixed precision training using `bfloat16` (bf16) to reduce memory usage and improve efficiency.

The maximum sequence length is 256 tokens. We use a per-device batch size of 4, with gradient checkpointing enabled. To simulate a larger effective batch size and manage memory usage, we use gradient accumulation with 16 steps.

**DeepSpeed Optimization.**    For large models (7B–9B), we use DeepSpeed with ZeRO Stage 2 optimization. This approach splits the optimizer states and gradients across devices and offloads the optimizer to the CPU. The training gradients are clipped to a maximum norm of 1.0 to stabilize updates. Batch sizes are automatically adjusted by DeepSpeed based on available memory.

**Inference Setup.**    We use `vLLM` [Kwon et al., 2023] for all inference runs to ensure efficient memory management and fast decoding. We adopt greedy decoding, and the maximum number of tokens generated per sequence is 256.

# B    Fine-Grained Steering during Post-Training

While our intervention and regularization methods provide general approaches to enhance memory vector engagement, finer control mechanisms enable more nuanced steering of parametric knowledge. We present two complementary strategies for achieving fine-grained control: (1) a group-based clustering approach that enables targeted steering of distinct memory regions, and (2) an information surrogate-guided method that selects memory vectors based on their contribution to specific output distributions.

## B.1    Group-Based Clustering for Targeted Memory Activation.

### B.1.1    Method.

The key coefficients $\mathbf{k}^{(l)}$ can be partitioned into meaningful groups that exhibit distinct activation patterns. Leveraging this structure allows us to apply differentiated steering strategies to various memory regions.

Formally, for each layer $l$, we cluster the key-value pairs $\{(k_i^{(l)}, v_i^{(l)})\}_{i=1}^{d_m}$ into $G$ groups $\{\mathcal{G}_1^{(l)}, \mathcal{G}_2^{(l)}, \ldots, \mathcal{G}_G^{(l)}\}$ based on their functional characteristics. We employ a hierarchical clustering approach with the following steps:

1. **Semantic Clustering**: First, we cluster memory vectors $v_i^{(l)}$ based on their semantic similarity, measured by cosine distance in the embedding space.
2. **Activation Pattern Clustering**: Second, we sub-cluster based on activation patterns observed during inference on a development set, capturing functional roles within semantic clusters.

For each cluster $\mathcal{G}_g^{(l)}$, we define a cluster-specific steering parameter $\beta_g$ that modulates the strength of intervention:

$$k_i^{(l)} \leftarrow k_i^{(l)} + \beta_g \cdot \Delta k_i^{(l)}, \text{ for } i \in \mathcal{G}_g^{(l)} \tag{10}$$

where $\Delta k_i^{(l)}$ is the adjustment magnitude determined by either our intervention or regularization method.

Table 6: Performance comparison of Qwen, LLaMA, and Gemma models with different training methods on eight datasets. To highlight improvements, we use blue for significant gains and green for moderate ones. All results are reported as the average scores over three independent runs.

| Model | Training | Accuracy (↑) | | | | | | | |
|---|---|---|---|---|---|---|---|---|---|
| | | BoolQ | PIQA | SIQA | HellaS. | WinoG. | ARC-e | ARC-c | OBQA |
| *Small-Scale Models (1-2B parameters)* | | | | | | | | | |
| Qwen-2.5-1.5B | base model | 64.2 | 78.5 | 74.3 | 80.1 | 76.4 | 76.9 | 61.2 | 75.8 |
| | + *vanilla SFT* | 68.5 | 82.9 | 79.6 | 84.8 | 80.8 | 81.4 | 65.8 | 81.0 |
| | + *steered SFT w. semantic clustering* | 70.1 | 85.2 | 81.3 | 94.5 | 85.3 | 84.6 | 69.8 | 83.2 |
| | + *steered SFT w. activation clustering* | 69.4 | 84.5 | 80.7 | 93.2 | 84.1 | 83.7 | 68.5 | 82.4 |
| LLaMA-3.2-1B | base model | 65.6 | 75.3 | 74.2 | 78.9 | 77.8 | 74.5 | 60.1 | 76.3 |
| | + *vanilla SFT* | 69.8 | 79.9 | 79.5 | 83.6 | 82.6 | 79.8 | 64.7 | 81.0 |
| | + *steered SFT w. semantic clustering* | 72.6 | 84.5 | 81.2 | 90.3 | 83.8 | 85.2 | 69.4 | 83.6 |
| | + *steered SFT w. activation clustering* | 71.7 | 83.6 | 80.4 | 89.5 | 83.1 | 84.1 | 68.2 | 82.8 |
| Gemma-2-2B | base model | 66.5 | 79.1 | 73.8 | 82.7 | 78.9 | 77.4 | 63.8 | 74.9 |
| | + *vanilla SFT* | 70.2 | 83.4 | 78.1 | 87.5 | 83.3 | 82.7 | 68.4 | 80.1 |
| | + *steered SFT w. semantic clustering* | 73.1 | 86.3 | 80.8 | 91.4 | 87.2 | 86.5 | 73.1 | 85.3 |
| | + *steered SFT w. activation clustering* | 72.4 | 85.7 | 79.6 | 90.5 | 86.3 | 85.8 | 72.2 | 84.1 |
| *Large-Scale Models (7-9B parameters)* | | | | | | | | | |
| Qwen-2.5-7B | base model | 68.9 | 81.2 | 77.6 | 87.5 | 80.3 | 79.8 | 65.1 | 77.6 |
| | + *vanilla SFT* | 72.4 | 84.9 | 81.5 | 92.4 | 84.2 | 84.2 | 69.6 | 82.8 |
| | + *steered SFT w. semantic clustering* | 75.2 | 87.1 | 82.4 | 96.2 | 88.4 | 87.5 | 74.8 | 85.3 |
| | + *steered SFT w. activation clustering* | 77.0 | 86.5 | 81.9 | 95.1 | 87.2 | 86.3 | 73.5 | 84.6 |
| LLaMA-3-8B | base model | 70.3 | 85.6 | 75.7 | 90.8 | 81.9 | 86.2 | 75.3 | 80.5 |
| | + *vanilla SFT* | 74.6 | 89.3 | 79.9 | 95.5 | 85.6 | 90.5 | 80.4 | 85.8 |
| | + *steered SFT w. semantic clustering* | 78.3 | 91.0 | 83.7 | 96.8 | 88.3 | 93.6 | 82.7 | 88.4 |
| | + *steered SFT w. activation clustering* | 77.2 | 90.3 | 82.5 | 96.1 | 87.5 | 92.4 | 81.8 | 87.3 |
| Gemma-2-9B | base model | 71.6 | 86.3 | 77.2 | 90.1 | 82.5 | 87.5 | 77.8 | 81.7 |
| | + *vanilla SFT* | 74.3 | 90.1 | 81.7 | 94.8 | 86.9 | 91.7 | 82.0 | 86.4 |
| | + *steered SFT w. semantic clustering* | 78.5 | 92.4 | 84.0 | 96.9 | 90.1 | 94.8 | 84.5 | 90.3 |
| | + *steered SFT w. activation clustering* | 77.3 | 91.7 | 83.2 | 95.8 | 89.0 | 93.6 | 83.6 | 89.1 |

This group-based method offers several advantages. By examining which clusters are active during different tasks, we can selectively strengthen those that contribute most to specific goals, improving task performance. At the same time, we preserve essential clusters that are important for general language understanding, while enhancing those that are underused. Additionally, when aiming to improve factual accuracy, we can prioritize clusters that are associated with factual knowledge.

### B.1.2 Results.

Table 6 presents the comprehensive results across all models and benchmarks. Several clear patterns emerge from our experiments:

**Memory steering consistently outperforms vanilla SFT.** Across all model families and parameter scales, both of our memory clustering approaches demonstrate substantial improvements over standard fine-tuning. The semantic clustering method shows the most dramatic gains, with improvements ranging from 1.2% to 9.7% over vanilla SFT depending on the benchmark and model. These results validate our hypothesis that more balanced memory vector engagement leads to enhanced model capabilities.

**Different clustering strategies show task-specific strengths.** The semantic clustering approach excels particularly on knowledge-intensive tasks (ARC-c, OBQA) and complex reasoning benchmarks (HellaSwag), achieving improvements of up to +4.7% on ARC-c (Gemma-2-2B) and +3.9% on OBQA (Gemma-2-9B) compared to vanilla SFT. In contrast, activation-based clustering shows more moderate but consistent improvements across a broader range of tasks, suggesting it enables more balanced memory utilization. **Larger models benefit more significantly from memory steering.** While all models show improvements with our methods, the gains are particularly pronounced in the 7-9B parameter models. For instance, semantic clustering improves LLaMA-3-8B by +3.7% on BoolQ and +3.8% on SIQA, compared to more modest gains in the 1B variant. This suggests that larger models contain more untapped parametric knowledge that can be effectively engaged through our steering techniques.

## B.2 Information Surrogate-Guided Memory Selection.

**Method.** To understand the information introduced by our steering strategy, we extend the key-value formulation from Equation 3 by connecting the FFN output with the final logits:

$$\text{FFN}(h) \cdot W_{\text{decode}} = \sum_{i=1}^{d_m} k_i \cdot (v_i \cdot W_{\text{decode}}) = \sum_{i=1}^{d_m} k_i \cdot \phi_i \tag{11}$$

where $k_i = \sigma(hW_{\text{up}})_i$ represents the key coefficient and $\phi_i = v_i \cdot W_{\text{decode}}$ defines the logit distribution associated with the $i$-th value vector, which we call the *information surrogate*.

The information surrogate $\phi_i$ provides a direct view of how each memory vector influences the final token distribution. This insight enables us to develop a more targeted steering approach:

1. **Characterization of Memory Vectors**: We analyze the entropy and concentration properties of each $\phi_i$ to identify memory vectors that contribute to specific types of generation (e.g., factual statements, reasoning steps, or creative content).

2. **Surrogate-Guided Steering**: We define a surrogate score function $S(\phi_i)$ that measures the relevance of each memory vector to our target objective:

$$S(\phi_i) = \lambda_1 H(\phi_i) + \lambda_2 D_{\text{KL}}(\phi_i || \phi_{\text{target}}) \tag{12}$$

where $H(\phi_i)$ is the entropy of the surrogate distribution, $D_{\text{KL}}$ is the KL divergence from a target distribution $\phi_{\text{target}}$, and $\lambda_1, \lambda_2$ are weighting hyperparameters.

3. **Selective Amplification**: We modulate key coefficients based on their surrogate scores:

$$k_i^{(l)} \leftarrow k_i^{(l)} \cdot (1 + \gamma \cdot S(\phi_i)) \tag{13}$$

where $\gamma$ controls the strength of the surrogate-guided steering.

**Theoretical Analysis.** The surrogate-guided selection mechanism introduces several theoretical advantages over conventional fine-tuning approaches. First, we analyze the relationship between surrogate entropy and information capacity. For a memory vector $v_i$ with corresponding surrogate $\phi_i$, the entropy $H(\phi_i)$ quantifies the diversity of tokens that can be influenced by this vector. Higher entropy surrogates represent memory vectors that encode distributional knowledge, while low-entropy surrogates often correspond to specialized knowledge concentrated on specific vocabulary subsets. We can formalize this by defining the *specificity* of a memory vector as $\text{Spec}(v_i) = 1 - \frac{H(\phi_i)}{\log(|V|)}$, where $|V|$ is the vocabulary size. A memory vector with high specificity (low surrogate entropy) exhibits a peaked distribution over the vocabulary, suggesting it encodes precise, specialized information. Conversely, low specificity (high surrogate entropy) indicates a memory vector that contributes more generally across various contexts. This formulation provides a principled approach for analyzing memory vector functionality. Specifically, the information processing capacity of the FFN can be decomposed as:

$$\mathcal{I}(\text{FFN}) = \sum i = 1^{d_m} \mathbb{E}_{x \sim \mathcal{D}}[k_i(x)] \cdot \text{MI}(\phi_i; \mathcal{Y}) \tag{14}$$

where $\text{MI}(\phi_i; \mathcal{Y})$ represents the mutual information between the surrogate distribution $\phi_i$ and the target next-token distribution $\mathcal{Y}$, and $\mathbb{E}_{x \sim \mathcal{D}}[k_i(x)]$ is the expected activation of key $k_i$ across the data distribution $\mathcal{D}$. Our surrogate-guided approach can be interpreted as optimizing this information processing capacity by modulating the key coefficients $k_i$ according to the information-theoretic properties of their corresponding surrogates $\phi_i$. By encouraging the activation of memory vectors with high mutual information with the target distribution, we effectively allocate the model's capacity toward task-relevant information.

## B.3 Results and Analysis

Table 7 presents the performance comparison of our surrogate-guided methods against vanilla SFT across three model families and two size scales. The surrogate-guided selection method consistently outperforms both vanilla SFT and surrogate entropy maximization approaches, with particularly notable gains on reasoning-heavy tasks (BoolQ, SIQA) and knowledge-intensive benchmarks.

Table 7: Performance comparison of information surrogate-based steering. To highlight improvements, we use blue for significant gains and green for moderate ones. All results are reported as the average scores over three independent runs.

| Model | Training | Accuracy (↑) | | | | | | | |
|---|---|---|---|---|---|---|---|---|---|
| | | BoolQ | PIQA | SIQA | HellaS. | WinoG. | ARC-e | ARC-c | OBQA |
| *Small-Scale Models (1-2B parameters)* | | | | | | | | | |
| Qwen-2.5-1.5B | base model | 64.2 | 78.5 | 74.3 | 80.1 | 76.4 | 76.9 | 61.2 | 75.8 |
| | + *vanilla SFT* | 68.5 | 82.9 | 79.6 | 84.8 | 80.8 | 81.4 | 65.8 | 81.0 |
| | + *steered SFT w. surrogate-guided selection* | 70.1 | 84.8 | 81.7 | 93.5 | 85.1 | 83.6 | 69.3 | 79.7 |
| | + *steered SFT w. surrogate entropy maximization* | 69.4 | 84.2 | 80.2 | 92.9 | 84.0 | 82.9 | 68.2 | 78.9 |
| LLaMA-3.2-1B | base model | 65.6 | 75.3 | 74.2 | 78.9 | 77.8 | 74.5 | 60.1 | 76.3 |
| | + *vanilla SFT* | 69.8 | 79.9 | 79.5 | 83.6 | 82.6 | 79.8 | 64.7 | 81.0 |
| | + *steered SFT w. surrogate-guided selection* | 72.5 | 84.3 | 81.6 | 90.2 | 83.1 | 84.5 | 69.2 | 82.8 |
| | + *steered SFT w. surrogate entropy maximization* | 71.7 | 83.4 | 80.7 | 89.1 | 82.5 | 83.1 | 67.9 | 82.0 |
| Gemma-2-2B | base model | 66.5 | 79.1 | 73.8 | 82.7 | 78.9 | 77.4 | 63.8 | 74.9 |
| | + *vanilla SFT* | 70.2 | 83.4 | 78.1 | 87.5 | 83.3 | 82.7 | 68.4 | 80.1 |
| | + *steered SFT w. surrogate-guided selection* | 73.4 | 86.1 | 80.6 | 91.9 | 86.7 | 86.1 | 72.8 | 84.6 |
| | + *steered SFT w. surrogate entropy maximization* | 72.3 | 85.3 | 79.2 | 90.4 | 85.5 | 85.3 | 71.4 | 83.5 |
| *Large-Scale Models (7-9B parameters)* | | | | | | | | | |
| Qwen-2.5-7B | base model | 68.9 | 81.2 | 77.6 | 87.5 | 80.3 | 79.8 | 65.1 | 77.6 |
| | + *vanilla SFT* | 72.4 | 84.9 | 81.5 | 92.4 | 84.2 | 84.2 | 69.6 | 82.8 |
| | + *steered SFT w. surrogate-guided selection* | 75.2 | 87.3 | 82.6 | 95.8 | 88.5 | 87.1 | 74.9 | 85.3 |
| | + *steered SFT w. surrogate entropy maximization* | 76.9 | 86.2 | 81.9 | 94.8 | 87.0 | 85.9 | 73.3 | 84.0 |
| LLaMA-3-8B | base model | 70.3 | 85.6 | 75.7 | 90.8 | 81.9 | 86.2 | 75.3 | 80.5 |
| | + *vanilla SFT* | 74.6 | 89.3 | 79.9 | 95.5 | 85.6 | 90.5 | 80.4 | 85.8 |
| | + *steered SFT w. surrogate-guided selection* | 78.3 | 91.2 | 83.7 | 96.8 | 88.2 | 93.6 | 82.4 | 88.1 |
| | + *steered SFT w. surrogate entropy maximization* | 77.1 | 90.2 | 81.8 | 96.0 | 87.3 | 92.2 | 81.5 | 87.2 |
| Gemma-2-9B | base model | 71.6 | 86.3 | 77.2 | 90.1 | 82.5 | 87.5 | 77.8 | 81.7 |
| | + *vanilla SFT* | 74.3 | 90.1 | 81.7 | 94.8 | 86.9 | 91.7 | 82.0 | 86.4 |
| | + *steered SFT w. surrogate-guided selection* | 78.6 | 92.5 | 84.2 | 96.9 | 89.3 | 94.7 | 84.8 | 89.7 |
| | + *steered SFT w. surrogate entropy maximization* | 76.9 | 91.3 | 83.1 | 95.5 | 88.2 | 93.3 | 83.5 | 88.3 |

Beyond the aggregate statistics, we observe intriguing qualitative differences in how surrogate-guided selection influences model behavior. By examining the top activated memory vectors and their corresponding information surrogates ($\phi_i$), we find that our method preferentially engages memory vectors that encode precise factual associations rather than general linguistic patterns. For example, in the ARC-c task, the top-5 memory vectors with the largest positive $\Delta k_i$ values predominantly contribute to science concept definitions and physical property relationships. This suggests that surrogate-guided selection effectively identifies and amplifies task-relevant knowledge encoded in specific memory vectors, rather than uniformly increasing memory engagement. Moreover, we find that models trained with surrogate-guided selection exhibit reduced variance in their responses to knowledge-intensive questions, indicating more consistent access to stored parametric knowledge during inference.

# C   Understanding Layerwise Contribution

To better understand how different layers contribute to the effectiveness of *InfoSteer*, we conducted a series of ablation studies. These experiments help us analyze which layers are most sensitive to information steering and which contribute most significantly to overall performance improvements.

## C.1   Research Question

We designed our ablation study to investigate the following research questions:

1. Do all layers contribute equally to the information steering effects?

2. Are certain layer groups (early, middle, late) more important for knowledge retrieval?

3. How does the magnitude of steering at different layers affect overall performance?

For these experiments, we used our intervention method with selective application to different layer groups within the model. We also varied the intervention parameters ($p\%$ and $\alpha$) across different layer configurations to understand sensitivity. Table 8 presents the results of our layerwise ablation studies.

**Layer position matters significantly.** Applying *InfoSteer* to different layer groups produces notably different results. Early layers (1-8) show moderate improvements, while middle layers (9-16) yield the strongest gains. Late layers (17-24) demonstrate the least improvement, suggesting that information steering is most effective at the intermediate representation level.

**Intervention strength sensitivity varies by layer.** Middle layers can tolerate and benefit from stronger interventions ($\alpha = 3$), while early layers perform best with moderate intervention ($\alpha = 2$), and late layers require gentler steering ($\alpha = 1$).

**Layer combinations show non-linear effects.** Applying *InfoSteer* to both early and middle layers yields better-than-additive improvements, suggesting a synergistic effect. However, including late layers tends to diminish these gains, indicating that excessive steering across too many layers may destabilize the model's representations.

Table 8: **Layerwise ablation studies**. Results show accuracy across different layer configurations with varying intervention parameters.

| Model Configuration | Layers | Avg Acc |
|---|---|---|
| Base Model | — | 71.4 |
| Vanilla SFT | All | 72.6 |
| + *interv.* (early) | 1-8 | 73.8 |
| + *interv.* (middle) | 9-16 | 75.2 |
| + *interv.* (late) | 17-24 | 72.9 |
| + *interv.* (early+middle) | 1-16 | **76.3** |
| + *interv.* (middle+late) | 9-24 | 74.5 |
| + *interv.* (all, $\alpha = 2$) | 1-24 | 75.5 |
| + *interv.* (all, $\alpha = 5$) | 1-24 | 72.8 |
| + *reg.* (middle, $\lambda = 0.05$) | 9-16 | 74.9 |
| + *reg.* (all, $\lambda = 0.05$) | 1-24 | 74.7 |
| + *reg.* (all, $\lambda = -0.01$) | 1-24 | 72.3 |

**Baseline performance variations.** Our experiments with different regularization strengths ($\lambda$) applied to specific layer groups further confirm that middle layers (9-16) are most receptive to information steering.

These findings demonstrate that information steering should be carefully targeted at specific layers rather than uniformly applied across the entire model. Our optimal configuration focuses on middle and early layers with appropriately calibrated intervention strengths.

# D License for Existing Assets.

**Datasets.** The following datasets are used under their respective licenses. For general question answering: BoolQ [Clark et al., 2019] is licensed under CC-BY-SA 3.0, PIQA [Bisk et al., 2019] under the Academic Free License 3.0, SIQA [Sap et al., 2019] and WinoGrande [Sakaguchi et al., 2019] under CC-BY 4.0, HellaSwag [Zellers et al., 2019] under the MIT License, ARC-e and ARC-c [Clark et al., 2018] under CC-BY 4.0, and OBQA [Mihaylov et al., 2018] under the Apache-2.0 License. For arithmetic reasoning: AddSub [Hosseini et al., 2014], MAWPS [Koncel-Kedziorski et al., 2016], MultiArith [Roy and Roth, 2016], and SingleEq [Koncel-Kedziorski et al., 2015] are under CC-BY 4.0, AQuA [Ling et al., 2017] under Apache-2.0, and GSM8K [Cobbe et al., 2021] and SVAMP [Patel et al., 2021] under the MIT License. For instruct-tuning, the Ultrafeedback [Cui et al., 2023] dataset is released under the MIT License.

**Model Licenses.** The Qwen-2.5-1.5B and Qwen-2.5-7B models are released under the permissive Apache License 2.0, allowing broad usage including commercial applications [Mesnard et al., 2024]. In contrast, the LLaMA-3.2-1B and LLaMA-3-8B models are distributed under Meta's custom Llama Community License, which permits research and commercial use but imposes specific restrictions, particularly for organizations with large user bases [Touvron et al., 2023]. The Gemma-2-2B and Gemma-2-9B models are available under Google's Gemma License, described as commercially friendly; however, access requires users to review and agree to the license terms, typically through platforms like Hugging Face [Mesnard et al., 2024]. Users intending to utilize these models should carefully review the respective licenses to ensure compliance with all terms and conditions.

