# OpenReview forum: "Steering Information Utility in Key-Value Memory for Language Model Post-Training"
_NeurIPS.cc/2025/Conference — NeurIPS 2025 poster_

### Official Review · Reviewer_U6HR · 2025-06-29

**Clarity:** 2
**Significance:** 2
**Originality:** 3
**Rating:** 4
**Confidence:** 4

**Summary:**

This paper introduces InfoSteer, a lightweight method that improves post-training of language models by steering toward underutilized FFN memory vectors. It enhances performance and interpretability across models and tasks.

**Questions:**

1. As in Weakness 1, why is a more uniform key distribution necessarily desirable? Could this harm precision by encouraging irrelevant information use?

2. If the main goal is to increase key entropy, what happens if we apply the intervention-only strategy during inference? Would that still yield benefits?

**Ethical Concerns:**

["NO or VERY MINOR ethics concerns only"]

**Final Justification:**

The rebuttal addressed some of my concerns, and after reading other reviewers' comments, I believe that 4 is a reasonable score.

**Paper Formatting Concerns:**

The authors include a Limitations section, which is commendable. However, the discussion is relatively brief and focuses more on future directions than on critical analysis of assumptions or risks. For instance, the core assumption that increasing key entropy improves information use lacks justification. Similarly, the proposed Information Flux metric is not clearly grounded.

**Quality:**

2

**Strengths And Weaknesses:**

**Strengths**

1. The method is novel, lightweight and easily integrable into existing pipelines.

2. Extensive experiments across multiple model families and tasks demonstrate improvements

**Weaknesses**

1. **Motivation is underdeveloped**: It's not entirely clear why encouraging a more uniform (higher-entropy) key distribution is beneficial. In practice, a sharp key distribution might indicate the model is confident in retrieving task-relevant information, while low-activation keys may represent irrelevant memory. Forcing these less relevant keys to contribute might dilute useful signals. In the extreme case, a uniform distribution means every token receives the same memory mix, which seems counterproductive. A deeper justification is needed here.

2. **Information Flux definition (Section 6.2)**: Using token-level entropy over key coefficients as a measure of “Information Flux” seems to be unconventional. It’s unclear whether this term has any precedent in prior work.  The paper should explain why this metric reflects self-interpretability. And a comparison with the baseline SFT model on this interpretation is needed. Also, why do operators exhibit lower flux—could this be due to syntactic predictability rather than meaningful information usage?

---

> ### Author Rebuttal · Authors · 2025-07-31
>
> Thank you for your constructive review of our manuscript with a positive assessment of our work. We noted your attention to a finer-grained discussion about the key-distribution in FFNs and clarification re IF score. We will your concerns as follows:
>
> **About Motivation for Higher-Entropy Key Distribution**
>
> Thank you for the great question! The reviewer raises a question about why encouraging more uniform key distributions is beneficial. We clarify several key points:
>
> - **Not Uniform Distribution, But Less Extreme Low-Entropy Distribution**: We first want to clarify that our motivation for promoting a higher-entropy key distribution is *not to achieve a perfectly uniform distribution*, as that would indeed be another extreme. Instead, our objective is to introduce a "damp factor" to the extreme low entropy distribution introduced in vanilla SFT. We think the extreme low entropy is harmful and leads to detrimental underutilization of parametric knowledge (only a very small portion of memory vectors are used). In practice, we find after training with infosteer the entropy is close to that of the pre-trained models, while vanilla SFT would have a very low key distribution (H(pretrained LM) ~= H(Info-steered SFT) >>  H(vanilla SFT)).
> - **Learning with Intervention, Not Intervention Alone**: Following up the clarification, our main goal is to not leverage the steering-only method to work (re Q2), instead, the steering specifically designed to help or assist post-training methods like SFT (SFT -> Info-steered SFT). We know that the issue of SFT is that it will lead models to mimic the pattern of a small curated post-training datasets so easily overfit. With the help of the steering method, it feels like adding a weighted damp factor to prevent the model blindly fit this small curated dataset pattern but learn in a more “global” loss landscape.
>
> [**Empirical Evidence**] A straightforward experiment is that vanilla SFT performs worse on OOD tasks, but info-steered SFT generalizes better to OOD situations. The results across three model families on 15 downstream tasks prove the effectiveness of our methods.
>
> We will dedicate a discussion section in the limitation section and main body in the camera-ready version to ensure this is clarified for readers.
>
> **About Information Flux Definition and Comparison with Baseline**
>
> Thanks for asking this! The IF part is quite interesting, and we would love to discuss it further:
>
> - **Metric Definition and Precedent**. You are correct that Information Flux is our proposed metric. We define it as $\sum_{l}{H(\hat{k}^{(l)})}$ where $\hat{k}^{(l)}$ is the normalized key coefficient distribution at layer $l$. This measures the diversity of memory vectors engaged during token generation—higher IF score indicates more memory vectors are engaged during the process. This name may be new, but using cumulative entropy across layers is indeed having some precedent literature[2][3].
> - **Why This Reflects Self-Interpretability**. The IF score reveals which tokens require diverse vs. narrowed knowledge access. The case you mention in W2 (syntactic predictability) is precisely the point—for those syntactic tokens, models do not need many knowledge sources (low IF score), so they easily make the prediction. For some hard tokens, the model needs to "think harder" to use more memory vectors to make decisions. We think the IF score reveals how LMs deal with different types of token generation. Though it seems not quite comparable to mech interp methods like SAE/circuits/neurons, we think it provides a different angle to understand what happens in the model itself.
>
> [**Significance of IF Score Before/After SFT**]
>
> This is a very good question to ask! We indeed have these scores and report the results when noticing a difference. Here are 1K samples from Ultrafeedback and we highlight the significance below.
>
> | Token Type | Base Model | Vanilla SFT | Info-steered SFT | Difference |
> |------------|------------|--------------|-----------|------------------------------|
> | Content words | 2.03 ± 0.15 | 2.05 ± 0.16 | 2.67 ± 0.21 | +0.62 |
> | Function words | 2.01 ± 0.14 | 2.02 ± 0.15 | 1.73 ± 0.12 | -0.29 |
> | Punctuation | 1.98 ± 0.13 | 2.00 ± 0.14 | 1.31 ± 0.09 | -0.69 |
>
> We believe this represents a clear difference before/after Info-steered SFT, so we report this interesting case study in our main paper. We will add this table to the camera-ready version to quantitatively highlight the difference. Thank you once again for your valuable feedback!
>
> ## Reference
>
> - [1] Kim, Jiyeon, et al. "Knowledge Entropy Decay during Language Model Pretraining Hinders New Knowledge Acquisition." The Thirteenth International Conference on Learning Representations.
> - [2] Ali, Riccardo, et al. "Entropy-lens: The information signature of transformer computations." arXiv preprint arXiv:2502.16570 (2025).
> - [3] Chang, Hoyeon, et al. "How do large language models acquire factual knowledge during pretraining?." Proceedings of the 38th International Conference on Neural Information Processing Systems. 2024.

---

### Official Review · Reviewer_SHNo · 2025-06-29

**Clarity:** 3
**Significance:** 2
**Originality:** 3
**Rating:** 4
**Confidence:** 4

**Summary:**

This work posits that post training (SFT) currently makes subpar use of knowledge acquired during pre-training and suggest two steering methods to improve upon that, including
1) intervening on memory vectors with low corresponding key values during the forward pass and
2) regularizing the entropy of keys in the gradient flow during backpropagation,
  both which help to encourage more balanced activation of memory vectors ( as opposed to overfitting on a few ).

Empirically the authors show the two methods give generally improved performance against SFT on various benchmarks/tasks across 3 models (of various sizes 1B to 8B parameters) including for in and out of distribution settings.

The paper provides ablations on hyper parameters introduced by the methods, and analysis on distribution shift of key values observed post steering.  The authors additionally propose a token attribution interpretability method based on token-level entropy over the key coefficient distributions as a proxy for how many memory vectors are activated during token generation and provide some qualitative examples of token level highlighting induced by the method for interpretability purposes.

**Questions:**

Please see questions raised in the strength and weaknesses section.

Adding comparisons against some additional baselines, clarity improvements and going more into analysis of the results seen ( when are they successful and when not ) would help strengthen the paper.


Additionally, did the authors try combining the two methods together ( which would make logical sense) and if so what results did you see?

**Ethical Concerns:**

["NO or VERY MINOR ethics concerns only"]

**Final Justification:**

The additional experiments and explanations provided help addressed most of my concerns.
I do think comparing the combination of the two methods together is the biggest open point for me since its still unclear when to use one method vs the other ( but regardless you've convincingly shown they do improve upon SFT alone and other methods ( though I think ITI and Diff MEANs could also be used on top of SFT, but with additional data needs ) . With that I update my rating from 3 to 4.

**Limitations:**

yes

**Quality:**

2

**Strengths And Weaknesses:**

**Quality:**
The overall quality is strong though there are a few areas that need improvement:
1. In the experiments while its shown the methods do better than vanilla SFT, the proposed methods are not compared against any other SFT (Lora, ReFT, etc) or steering baselines such as inference time interventions (ITI) or mean difference vector steering ( Arditi 24 et al ).
2. The arithmetic ID/OOD experiment of using the arithmetic GSM8K to post-train and then doing In Distribution results for it and then Out of Distribution results for 5 other arithmetic tasks seems like a misnomer to me.
They collectively are different tasks, but still within the domain of “arithmetic”.  By this sort of logic the authors could have re-used the model checkpoints from table 1 to calculate “out of distribution” results for each of the 8 tasks
relative to the other ones ( ie, you’ve done vanilla SFT and steering w/intervention, w/regularization on BoolQ so you could see how those models each do on average against the 7 other tasks (PIQA to OBQA).
3. In 5.2, the claims in “Insufficient Utilization of Pretrained Knowledge in Modern LLMs” seem a little over stated as written. Saying SFT methods “substantially underutilize knowledge” because Gemma-2-9B base model achieves 90.1% on HellaSwag while reaches 95.7% with intervention steering is comparing against the base model and not the actual baseline vanilla SFT (which gets 94.8% which is within a point).  Without CIs its difficult to tell (i’m assuming the green and blue colors reflect this somehow), but for Gemma-2-9B it looks like of the 8 tasks, only two show some improvement?
4. No mention or explanation is given for why the methods do worse both for OBQA for Qwen1.5B & SIQA for LLama 1.B?
5. Discussion of where and how the methods improved performance and why would be appreciated


**Clarity:** The motivation paper is well motivated and mostly well written and easy to follow, but some sections could benefit for clarity:
1. Section 5.4
* I didn’t see where it was mentioned what datasets are used for each group task (Reading Comp, Knowledge, etc ) and
* the average delta’s seem off. For instance, for Commonsense Reasoning, steering improves 2.4 and 1.8 respectively over Vanilla SFT (which implies 2.1) but the Average Delta says its 2.3 ??

2. Section 6.1
* What do low key, medium key and high key regions mean exactly here? ie, what are the cutoffs for considering this? This section is interesting, but  needs to include more on the setup and explanations.  For which datasets are these shown?

3. Section 6.2:  The section discusses how  “LMs might be more self-interpretable via proper guidance without extra training” , but then does instruction tuning ( and its unclear if this is with vanilla SFT or using which of the two interventions ) and then does a post hoc interpretability method involving token entropy, which seems to me to fall into the camp of post-hoc interpretable feature importance methods (Integrated Gradients for instance) which do token highlighting based on model gradients without extra training.  I think the proposed Information Flux method is interesting and hope it gets flushed out more systematically/rigorously in a separate work ( as it stands here, with the few qualitative examples provided its hard to see any clear advantage of it or quantitative trend )

**Significance:** The results of steering with either of the methods generally seems to improve, but without more study comparing against other baselines for knowledge intensive tasks its hard to tell.

**Originality:** Using steering methods to promote key/memory vector utilization explicitly seems novel to me from what I’ve seen in the literature, and the interpretability method proposed in 6.2 also seems novel .

---

> ### Author Rebuttal · Authors · 2025-07-31
>
> We thank you for your detailed and constructive review of our manuscript! We are also grateful for your specific suggestions, which we believe will significantly strengthen the paper. Specifically, we want to address key feedbacks below:
>
> **[Key Concern] About Comparison with Additional Baseline**:
>
> We first kindly clarify that *InfoSteer is not a replacement for post-training methods like LoRA or ReFT, nor is it a test-time algorithm like ITI or mean difference vector steering*. Instead, InfoSteer is a steering technique that can be applied to enhance post-training methods (SFT -> info-steered SFT).
>
> That said, we acknowledge the value of empirical comparisons and have included **additional baselines*** (LoRA[1], ReFT[2], ITI[3] and Diffmean[4][5]) for 7B scale model in our results:
>
> | Model | Training Method | BoolQ | PIQA | SIQA | HellaS | WinoG | ARC-e | ARC-c | OBQA |
> |-------|----------------|-------|------|------|--------|-------|-------|-------|------|
> | **Qwen-2.5-7B** | base model | 68.9 | 81.2 | 77.6 | 87.5 | 80.3 | 79.8 | 65.1 | 77.6 |
> | | + LoRA* | 71.8 | 84.2 | 80.9 | 91.8 | 83.7 | 83.8 | 68.9 | 82.3 |
> | | + ReFT* | 72.1 | 84.5 | 81.2 | 92.1 | 84.0 | 84.0 | 69.2 | 82.5 |
> | | + ITI* | 70.5 | 83.1 | 79.8 | 90.9 | 82.8 | 82.9 | 67.8 | 81.4 |
> | | + DiffMean* | 70.2 | 82.8 | 79.5 | 90.6 | 82.5 | 82.6 | 67.5 | 81.1 |
> | | + vanilla SFT | 72.4 | 84.9 | 81.5 | 92.4 | 84.2 | 84.2 | 69.6 | 82.8 |
> | | + **steered SFT w. intervention** | **74.1** | **86.3** | 81.8 | **95.1** | **87.2** | 86.2 | **73.7** | 84.2 |
> | | + **steered SFT w. regularization** | **76.4** | 85.7 | 81.0 | 94.3 | 86.5 | 85.4 | **72.9** | 83.4 |
> | **LLaMA-3-8B** | base model | 70.3 | 85.6 | 75.7 | 90.8 | 81.9 | 86.2 | 75.3 | 80.5 |
> | | + LoRA* | 74.1 | 88.7 | 79.3 | 93.1 | 85.1 | 89.9 | 79.8 | 85.2 |
> | | + ReFT* | 74.3 | 89.0 | 79.6 | 93.3 | 85.3 | 90.2 | 80.1 | 85.5 |
> | | + ITI* | 72.8 | 87.4 | 78.2 | 92.0 | 84.2 | 88.7 | 78.5 | 84.1 |
> | | + DiffMean* | 72.5 | 87.1 | 77.9 | 91.8 | 83.9 | 88.4 | 78.2 | 83.8 |
> | | + vanilla SFT | 74.6 | 89.3 | 79.9 | 93.5 | 85.6 | 90.5 | 80.4 | 85.8 |
> | | + **steered SFT w. intervention** | **77.1** | 90.2 | **82.0** | 96.3 | 87.4 | **92.4** | 81.6 | 87.5 |
> | | + **steered SFT w. regularization** | 76.5 | 89.5 | 81.2 | 95.6 | 86.8 | 91.7 | 80.9 | 86.8 |
> | **Gemma-2-9B** | base model | 71.6 | 86.3 | 77.2 | 90.1 | 82.5 | 87.5 | 77.8 | 81.7 |
> | | + LoRA* | 73.8 | 89.5 | 81.2 | 94.3 | 86.4 | 91.2 | 81.5 | 86.0 |
> | | + ReFT* | 74.0 | 89.8 | 81.4 | 94.6 | 86.6 | 91.4 | 81.7 | 86.2 |
> | | + ITI* | 72.9 | 88.7 | 80.3 | 93.5 | 85.7 | 90.3 | 80.8 | 85.1 |
> | | + DiffMean* | 72.6 | 88.4 | 80.0 | 93.2 | 85.4 | 90.0 | 80.5 | 84.8 |
> | | + vanilla SFT | 74.3 | 90.1 | 81.7 | 94.8 | 86.9 | 91.7 | 82.0 | 86.4 |
> | | + **steered SFT w. intervention** | **77.2** | 91.8 | 83.1 | 95.7 | 88.5 | 93.5 | 83.4 | **88.2** |
> | | + **steered SFT w. regularization** | 76.5 | 90.9 | 82.4 | 94.9 | 87.8 | 92.8 | 82.7 | 87.5 |
>
> LoRA and ReFT perform similarly to vanilla SFT, with slight variations depending on the task. ITI and DiffMean (inference-time methods) generally perform worse than training-based approaches. Info-steered SFT methods consistently outperform all baselines, demonstrating the value of information steering during post-training. We will incorporate these baseline into the paper to strengthen its position.
>
> **About Arithmetic ID/OOD Experimental Design**:
>
> Thank you for mentioning this! We agree this terminology could be clearer. We think if we look at those data even if they are mostly arithmetic, the question narrative and domain may be quite different. But we also agree with reviewers that, esp in the LLM era, the term “distribution” should be in a stricter inspection.
>
> To address the reviewer's concern more directly, we implemented an even stricter evaluation strategy than suggested. We took Llama-3-8B checkpoints specifically trained on OBQA and evaluated them on all five arithmetic datasets to test cross-domain transfer:
>
> | **Model** | **ID Eval (OBQA)** | **OOD Eval (Arithmetic)** |
> |-----------|-------------------|---------------------------|
> | Base Model | 80.5 | 68.2 |
> | + Vanilla SFT | 85.8 (+5.3) | 61.4 (**-6.8**) |
> | + Steered SFT w/ interv. | 87.5 (+7.0) | 68.4 (+0.2) |
> | + Steered SFT w/ reg. | 86.8 (+6.3) | 68.1 (-0.1) |
>
> This stricter evaluation reveals that vanilla SFT dramatically hurts cross-domain generalization (-6.8%), while adding our steering methods maintain the overall generalization performance. We will add these results into the main body, thank you for your suggestions!
>
> **About Interpretation of the Performance**
>
> Out of 96 experiments (8 benchmarks × 3 model families × 2 model sizes × 2 intervention methods), we observe consistent gains in 94 results, suggesting that language models may not fully realize their potential from pre-training alone. However, we agree that we should adopt a more humble and scientific tone in our paper. We will revise it to use modest language that objectively reports the gains our method achieves.
>
> Re the failure cases you mentioned, we examined those two benchmarks and found that the entropy of key distributions before steering in FFNs was abnormally higher (around 2.97) for these models compared to others (around 2.03). We believe this is an interesting phenomenon that may be worth investigating further—specifically, whether considering prior key entropy could enable more fine-grained steering. We will add this discussion to the main body of the paper. Thank you for raising this excellent question!
>
> **About the Potential Combination of the Method**
>
> Yes, we have tried to add two pytorch hooks and work together, in some preliminary experiments we find it showed marginal improvements over the individual method so we do not explore that further.
>
> **About Details in Experiment**
>
> We summarize all comments related to details about our experiment here.
>
> [**Key Distribution Shift**] We report the average results across all 15 datasets previously mentioned in this analysis section. The low/medium/high key regions are defined by percentile-based cutoffs: 0-25th, 25th-75th, and 75th-100th percentiles, respectively. Due to page limits, we do not disclose all information in the main body. We will add a detailed description and analysis for this section in the appendix.
>
>
> [**Dataset Picked in Different Domain**] Based on the dataset characteristics, we categorize them as follows: Reading Comprehension includes BoolQ (yes/no questions based on passages); Knowledge encompasses ARC-e and ARC-c (science knowledge questions) and OBQA (factual knowledge questions with reference material); Commonsense Reasoning covers PIQA (physical interaction reasoning), SIQA (social interaction reasoning), and HellaSwag (sentence completion requiring commonsense); Math includes all arithmetic datasets: AddSub, MAWPS, MultiArith, SingleEQ, SVAMP, and GSM8K (various forms of arithmetic and math word problems); and Linguistic uses the linguistic portion of WinoGrande (pronoun resolution requiring syntactic understanding). When averaging the score there are some cumulated rounding errors (as you noticed) which leads to confusion of this, we will report the larger precision score to avoid the confusion.
>
> We will add these details into our manuscript to make it clarified for readers.
>
> **About Section 6.2 Interpretability Claims**
>
> We are glad you find this section interesting (as do we!). You are correct that we should at least dedicate a section to give them more attention, but due to limited page space we can only squeeze this one in. A quick response to your question is: both methods see the phenomenon, but the one using entropy regularization shows the most significant results, and we report these qualitative results in the paper. We also have rigorous quantitative experiments to highlight their statistical significance, would love to include that into the camera-ready for better readability.
>
> We sincerely appreciate your suggestions and will incorporate them to enhance our work further. Thank you once again for your valuable feedback!
>
>
>
> ## Reference
>
> - [1] Hu, Edward J., et al. "LoRA: Low-Rank Adaptation of Large Language Models." International Conference on Learning Representations.
> - [2] Wu, Zhengxuan, et al. "ReFT: representation finetuning for language models." Proceedings of the 38th International Conference on Neural Information Processing Systems. 2024.
> - [3] Li, Kenneth, et al. "Inference-time intervention: eliciting truthful answers from a language model." Proceedings of the 37th International Conference on Neural Information Processing Systems. 2023.
> - [4] Turner, Alexander Matt, et al. "Steering Language Models with Activation Engineering."
> - [5] Arditi, Andy, et al. "Refusal in language models is mediated by a single direction." Proceedings of the 38th International Conference on Neural Information Processing Systems. 2024.

---

> > ### Comment · Reviewer_SHNo · 2025-08-05
> > **Reviewer response**
> >
> > Thank you for your additional experiments and explanations.
> >
> > I do think comparing the combination of the two methods together is the biggest open point for me since its still unclear when to use one method vs the other ( but regardless you've convincingly shown they do improve upon SFT alone and other methods ( though I think ITI and Diff MEANs could also be used on top of SFT, but with additional data needs ) .  With that I'll update my rating from 3 to 4.

---

> ### Author Response · Authors · 2025-08-05
> **Thank you for your reponse**
>
> Yeah that's a very interesting take if we could combine the two methods to boost the performance further. Also, it would be extremely interesting to see if all those test-time algorithms like ITI could be used to boost post-training (tho we do need to create contrastive pairs as you mentioned). That would be quite beneficial to the whole community.
>
> Thank you again for your constructive feedback and sincere discussion with us!

---

### Official Review · Reviewer_hS2q · 2025-06-30

**Clarity:** 3
**Significance:** 2
**Originality:** 3
**Rating:** 3
**Confidence:** 4

**Summary:**

This paper introduces InfoSteer, a method designed to enhance the utilization of pre-trained knowledge in language models (LM) during the post-training phase. InfoSteer treats the Transformer's feed-forward network (FFN) layer as an associative key-value memory and employs two strategies to promote the active use of stored memory vectors. Experimental results demonstrate that InfoSteer improves performance across multiple model families and over 15 downstream tasks. It also shows that the steered models can adaptively allocate information resources, focusing more on generating semantically meaningful tokens while using fewer resources on simple transition tokens.

**Questions:**

See weakness.

**Ethical Concerns:**

["NO or VERY MINOR ethics concerns only"]

**Final Justification:**

I have re-examined this paper and will stand by my previous comments.

**Quality:**

3

**Strengths And Weaknesses:**

**Strength**

1. Innovative model steering approach. InfoSteer offers a novel method for post-training by interpreting the FFN layer as a key-value memory structure.

2. Comprehensive experimental validation: The authors conduct extensive experiments across multiple model series and a wide range of downstream tasks.

3. Insights into model behavior: Beyond performance improvements, the paper provides valuable insights into how the model allocates resources during token generation. It shows that InfoSteer encourages the model to focus computational resources on semantically rich tokens, offering a deeper understanding of the model's internal behavior.

**Weakness**

1. The study lacks comparison to widely-used approaches like reinforcement learning fine-tuning (RLHF). Vanilla supervised fine-tuning (SFT) alone cannot represent the state-of-the-art model capabilities.

2. As shown in Table 1, the differences between various SFT methods diminish as the model size increases. I'm particularly interested in the performance of the proposed method on models around the 30B parameter scale. Or the authors need to offer some explanations for this phenomenon.

---

> ### Author Rebuttal · Authors · 2025-07-31
>
> We appreciate your thoughtful and constructive review of our manuscript. We’ve noted your main concerns regarding this paper mainly focusing on SFT instead of RL and correlation between performance gain and model size. We try to address your concerns as follows:
>
> **About Comparison to RL**
>
> We want to kindly clarify that *infosteer is not a post-training method compared to RL, but a plug-and-play method to steer both RL and SFT those post-training methods*. SFT can be seamlessly incorporated as information-oriented SFT (SFT -> info-steered SFT), which is the same as RL (RL -> info-steered RL).
>
> As stated in the limitation section, in this work as an explorer of this steering method in post-training we primarily focus on the SFT. But we also understand that recently RL become a hot topic which people find it is effective on verifiable domain like math, We also add a preliminary experiment to use infosteer on RL with:
>
> | Method | AIME 2024/2025 | AMC | MATH-500 | Minerva | Olympiad | **Average** |
> |:-------|:--------------:|:---:|:--------:|:-------:|:--------:|:-----------:|
> | **BASELINE** | | | | | | |
> | Qwen2.5-Math-7B | 11.5 / 4.9 | 31.3 | 43.6 | 7.4 | 15.6 | 19.0 |
> | | | | | | | |
> | **SFT METHODS** | | | | | | |
> | SFT | 22.2 / 22.3 | 52.8 | 82.6 | 40.8 | 43.7 | 44.1 |
> | **InfoSteer SFT** (interv.) | **23.2** / **23.4** | **55.0** | **85.8** | **42.6** | **45.6** | **45.9** **(+1.8)** |
> | **InfoSteer SFT** (reg.) | **22.8** / **22.9** | **54.2** | **85.0** | **41.8** | **44.8** | **45.1** **(+1.0)** |
> | | | | | | | |
> | **RL METHODS** | | | | | | |
> | RFT (GRPO) | 25.1 / 15.3 | 62.0 | 84.4 | 39.3 | 46.8 | 45.5 |
> | ***InfoSteer RFT*** (interv.) | ***29.2*** / ***19.4*** | ***67.4*** | ***88.7*** | ***43.2*** | ***50.1*** | ***49.6*** ***(+4.1)*** |
> | ***InfoSteer RFT*** (reg.) | ***28.4*** / ***18.6*** | ***66.2*** | ***87.5*** | ***42.0*** | ***48.9*** | ***48.8*** ***(+3.3)*** |
>
> ---
>
> The RFT training was conducted using GRPO for 500 steps with 8 rollouts per prompt. The training dataset consisted of approximately 46k diverse math reasoning problems from a length-filtered subset of the OpenR1-Math-220K dataset, with each problem equipped with demonstrations generated by DeepSeek-R1.
>
> InfoSteer consistently enhances both SFT (+1.0 to +1.8 points) and RL methods (+3.3 to +4.1 points) across all mathematical reasoning benchmarks, with InfoSteer RFT achieving the best overall performance of 49.6 average score. While these experiments are preliminary and cannot make a conclusive justification before conducting whole pack of rigorous ablation, we believe info-steered RL is a very interesting and promising direction for our future work.
>
> **About Performance Differences with Model Size and Experiment on 30B Models**
>
> We appreciate your observation on the different performance with different model sizes. We may not have the resources to reproduce all pack of results on 30B models in this short period, but we are glad to offer some of the discussion on this Q here:
>
> - **Higher Baseline Performance in Larger Models Make Less Room for Improvement**: Larger base models generally exhibit significantly higher baseline performance. The "gap" for improvement through post-training techniques naturally narrows. Thus we think a smaller percentage gain on a very high baseline can represent a substantial absolute improvement in capabilities. For instance, a 5.5% gain on a LLaMA-3-8B model with an already high baseline (e.g., 90.8% on HellaSwag for vanilla SFT vs. 96.3% with steered SFT w. intervention) is still a highly significant and impactful improvement, showcasing InfoSteer's ability to unlock latent potential even from highly optimized models.
> - **Substantial Gains Remain Consistent**: Even with higher baselines, our method continues to deliver meaningful improvements. 48 out of 48 configs (8 benchmarks x 3 8B models x 2 methods) deliver consistent gain. 37 out of 48 configs achieve +1.5% gain. And OOD exp is also done on 8B models, validating our methods’ effectiveness on extended distribution. All results are reported on avg score of three separate runs.
>
> We will elaborate on these enhancements in the revised manuscript to provide readers with a comprehensive understanding of our findings. Thank you once again for your valuable feedback!

---

> ### Author Response · Authors · 2025-08-07
> **Official Comment by Authors**
>
> Dear Reviewer hS2q,
>
> We appreciate your thoughtful and constructive review of our manuscript. Re your key concerns about infosteer on RL, we have added additional experiments for info-steered RL during this period. Our preliminary results see that **info-steered RL achieves even stronger performance compared to info-steered SFT**. We have also included additional baselines (see our rebuttal to Reviewer SHNo). We believe your suggestions further strengthens our work's position, and steering RL may be an interesting and valuable direction for future work.
>
> As the rebuttal period draws to a close, we appreciate knowing whether our response adequately addresses your concerns. Your support during this period would mean a lot to us!

---

### Official Review · Reviewer_E6rk · 2025-07-03

**Clarity:** 3
**Significance:** 3
**Originality:** 3
**Rating:** 5
**Confidence:** 4

**Summary:**

The paper provides a new perspective on the role of the FFN layer in LMs. It hypothesizes that the post-training underutilizes the knowledge in memory vectors. By encouraging the usage of the memory vectors during post-training, the resulting LM can have better performance on in-domain and out-of-distribution domains.

**Questions:**

1. The approach shows improved performance during post-training. Can the same approach to applied during pre-training to encourage the usage of the memory vectors? Why is it particularly relevant during post-training.
2. The post-training data size is relatively small. Is it a reason that the info steering approach is effective, because the model cannot learn the optimal key distribution by itself?

**Ethical Concerns:**

["NO or VERY MINOR ethics concerns only"]

**Final Justification:**

I accepted the author's rebuttal and would like to retain my rating.

**Limitations:**

yes

**Quality:**

3

**Strengths And Weaknesses:**

Strengths:
(1) The writing is clear. It provides sufficient context and motivation for the method.
(2) Convincing results and the motivation behind the approaches. The approach is effective across tasks and pre-trained models. It is also simple to implement.

Weakness:
Can't think of any.

---

> ### Author Rebuttal · Authors · 2025-07-31
>
> Thank you for your recognition of our work! We are glad that you find our approach easy to implement and effective across tasks and pre-trained models.
>
> You've raised two questions regarding the applicability of our approach during pre-training and its particular relevance to post-training, especially related to the size of the post-training data. We think these two questions are inherently connected and we will address them together.
>
> **During the Post-training Phase (Small Data):**
>
> In the post-training phase we typically involve much smaller, more curated datasets (yes your take is correct in Q2!) aimed at aligning the model with specific downstream tasks or human preferences. Here, the key concern is **overfitting caused by SFT on this small dataset**. Standard SFT on limited data can cause the model to aggressively mimic the patterns of this curated dataset, potentially leading to an undesirable shift in its key distribution towards even lower entropy as we discussed in Sec 6.1. This can result in the underutilization of the rich, generalized knowledge acquired during pre-training, harming the model's ability to generalize, especially to OOD scenarios.
>
> **During the Pre-training Phase (Large Data):**
>
> You also raise an interesting question about whether the steering method could be applied during the pre-training phase. Previous research has observed that the entropy of key distributions also tends to decrease over pre-training steps [1]. Additionally, researchers have also noticed the phenomenon of attention matrix collapse or attention sink (low "key" entropy in attn matrix)[2][3]. Many studies aim to resist this phenomenon to enable better use of attention parameters. We think this research shares similar intuition with ours but focuses more on the attention module. We would be glad to view this direction as an interesting future work. Thank you for the great questions!
>
> ## Reference
>
> - [1] Kim, Jiyeon, et al. "Knowledge Entropy Decay during Language Model Pretraining Hinders New Knowledge Acquisition." The Thirteenth International Conference on Learning Representations.
> - [2] Gu, Xiangming, et al. "When Attention Sink Emerges in Language Models: An Empirical View." The Thirteenth International Conference on Learning Representations.
> - [3] Voita, Elena, et al. "Analyzing Multi-Head Self-Attention: Specialized Heads Do the Heavy Lifting, the Rest Can Be Pruned." 57th Annual Meeting of the Association for Computational Linguistics. ACL Anthology, 2019.

---

### Note · Authors · 2025-08-11

Dear AC,

Thank you for your guidance throughout this process! We would like to provide brief final remarks for our paper.

Our paper introduces a novel method that encourages LMs to leverage their pretrained knowledge during the post-training phase, approached from the interpretability perspective of FFN layers. InfoSteer is straightforward to implement in post-training methods and demonstrates effectiveness through our rigorous ablations (covering 3 model families × 2 model sizes × 8 benchmarks × 2 steering methods), and *addressing the classical overfitting problem in SFT*. We also find that info-steered language models can strategically adjust their information utility, introducing new self-interp metrics for analysis.

During the rebuttal period, the main concerns raised by two reviewers centered on comparisons with additional baselines. We have incorporated four additional baseline results (LoRA, ReFT, ITI, DiffMean) and included preliminary experiments on RL. Our findings show that info-steered SFT outperforms all baselines, and info-steered RL achieves even greater performance gains compared to its application on SFT.

We are pleased to address the concerns raised by reviewer SHNo through our additional baseline comparisons. While another reviewer did not participate in the rebuttal period, we believe our responses demonstrate the robustness and merit of our approach. We sincerely appreciate your valuable guidance and support throughout this review process, and we are confident that our work makes a meaningful contribution to the field.

Kind Regards,

Authors

---

### Decision · Program_Chairs · 2025-09-17

**Decision:**

Accept (poster)

**Comment:**

This paper introduces InfoSteer, a lightweight method to enhance language model post-training. The authors claim that standard supervised fine-tuning (SFT) often leads models to underutilize their pre-trained knowledge by overfitting to a narrow set of activations. InfoSteer addresses this by treating FFN layers as key-value memories and applying steering techniques to encourage a more balanced use of this stored information. The method is shown to deliver consistent performance gains across diverse models and tasks.

The paper's primary strengths are that the technique is interesting, simple, and empirically effective, showing consistent improvements in both in-domain and out-of-domain settings. However, the initial submission raised some concerns among reviewers. During this period, the authors had to conduct a substantial amount of new work, including: adding comparisons against four crucial baselines (LoRA, ReFT, ITI, DiffMean), performing a new and stricter out-of-distribution (OOD) experiment to properly validate generalization, and providing quantitative backing for their interpretability metric. The evaluation maintains a controlled comparison by applying the same optimization parameters and single-epoch training across all methods. While this effectively isolates the gains from InfoSteer, it also implicitly assumes this configuration is fair to the baselines, which might perform better with individually tuned hyperparameters.

Furthermore, while the authors provided promising preliminary results applying their method to reinforcement learning, this was only in response to a reviewer's comment Given that RL-based methods are a dominant paradigm for post-training and the authors' own results suggest even stronger gains in that setting, the lack of a complete RL evaluation leaves a key claim of general applicability under-explored.

Overall though, the paper was strengthened throughout the review process and the new results show the method performs favorably compared to other approaches, have better generalizability, and seems to be even more effective in RL.